# Hairpin trimer transition state of amyloid fibril

Levent Sari ®[1,2], Sofia Bali ®[3,4], Lukasz A. Joachimiak ®[4,5] & Milo M. Lin ®[1,2,4,6] ✉

Protein fibril self-assembly is a universal transition implicated in neurodegenerative diseases. Although fibril structure/growth are well characterized, fibril nucleation is poorly understood. Here, we use a computational-experimental approach to resolve fibril nucleation. We show that monomer hairpin content quantified from molecular dynamics simulations is predictive of experimental fibril formation kinetics across a tau motif mutant library. Hairpin trimers are predicted to be fibril transition states; one hairpin spontaneously converts into the cross-beta conformation, templating subsequent fibril growth. We designed a disulfide-linked dimer mimicking the transition state that catalyzes fibril formation, measured by ThT fluorescence and TEM, of wild-type motif - which does not normally fibrillize. A dimer compatible with extended conformations but not the transition-state fails to nucleate fibril at any concentration. Tau repeat domain simulations show how long-range interactions sequester this motif in a mutation-dependent manner. This work implies that different fibril morphologies could arise from disease-dependent hairpin seeding from different loci.

Amyloid fibrils are self-assembled protein complexes in which protein monomers interact in a stereotypical cross-beta structure: intermolecular backbone hydrogen bonding to form parallel beta sheets and intramolecular side-chain packing. Fibrils correspond to a universal solid phase of protein structure: experimentally determined structures of ordered aggregates across a wide range of proteins all sharing the cross-beta structure[1]. Fibrils also play a critical, yet not fully understood, role in neurodegenerative disorders like Alzheimer's Disease (AD) and Parkinson's disease. Under the right conditions, even typically non-aggregating proteins[2] can be induced to form cross-beta fibrils, leading to the hypothesis that the amyloid fibril is the universal thermodynamically favored state at sufficiently high protein concentration[1]. Recently, this empirical consensus was confirmed by a statistical mechanical model showing the thermodynamic preference of an ordered state over a disordered polymer melt under physiological conditions regardless of protein sequence[3]. Therefore, kinetics

rather than thermodynamics of fibril formation is the determining factor for predicting and controlling aggregation fate within the relevant physiological timescale.

The kinetic process of amyloid formation occurs in two consecutive stages: The first "lag-phase" stage, during which no fibrils are observed, and the second fibril "elongation/growth phase". Fibril growth during the second stage, involving monomer attachment to existing fibrils, is well understood[4,5]. As in the kinetic theory of chemical reactions[6], the first lag phase contains the rate-limiting primary nucleation process that proceeds via formation of the transition state complex[7,8], although secondary nucleation may dominate the lag phase if fibril fragmentation is prevalent[9]. Yet, little is known about the structural nature of the lag phase because the heterogeneity and transience of the transition state preclude atomistic experimental characterization. There are two transition state hypotheses (Fig. 1a): (i) a disordered transition state in which

[1]Green Center for Systems Biology, University of Texas Southwestern Medical Center, Dallas, TX, USA. [2]Lyda Hill Department of Bioinformatics, University of Texas Southwestern Medical Center, Dallas, TX, USA. [3]Molecular Biophysics Graduate Program, University of Texas Southwestern Medical Center, Dallas, TX, USA. [4]Center for Alzheimer's and Neurodegenerative Diseases, Peter O'Donnell Jr. Brain Institute, University of Texas Southwestern Medical Center, Dallas, TX, USA. [5]Department of Biochemistry, University of Texas Southwestern Medical Center, Dallas, TX, USA. [6]Department of Biophysics, University of Texas Southwestern Medical Center, Dallas, TX, USA. ✉e-mail: Milo.Lin@UTSouthwestern.edu

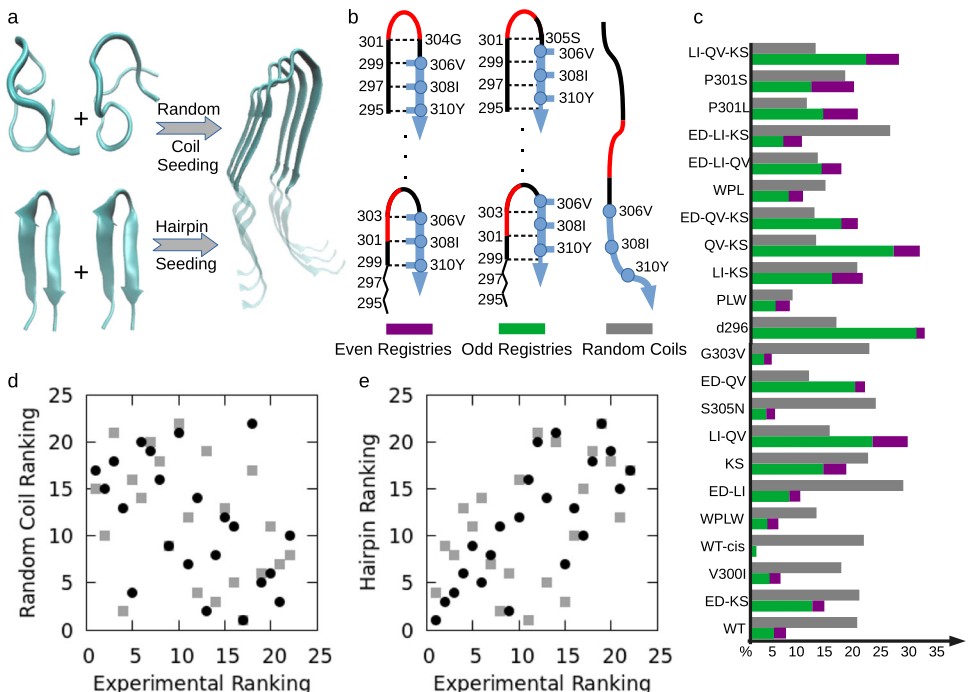

**Fig. 1 | Correlating fibril nucleation rates to structural motifs of the monomer ensemble across tau-derived mutant peptide library. a** Alternative hypotheses of amyloid fibril nucleation from random coil (disordered) versus hairpin (ordered) structural motifs. **b** Register of even (left) and odd (middle) hairpin pairings for tau 295–311, with their ensemble fraction represented in purple and green, respectively. Random coil ensemble fraction (right) is represented in gray. VQIVYK hexapeptide PHF6 motif, PGGG motif, and N terminal segment are shown in blue, red, and black, respectively. Note that, in odd hairpins, hydrophobic/aromatic even numbered residues (306V, 308I, and 310Y) have solvent exposed backbone

oxygens and amide hydrogens. **c** Calculated even (green) and odd (purple) hairpin probabilities and random coil probabilites (gray) as obtained from REMD/Amber96 atomistic simulations. **d** Anti-correlation between random coil contents and the experimental aggregation speeds [r = −0.37 and r = −0.47, respectively, by REMD/Amber96 (gray squares) and MD/Amber99sb-ildn (black circles)]. **e** Positive correlation between hairpin contents and experimental aggregation speeds [r = 0.47 and r = 0.76 respectively by REMD/Amber96 (gray squares) and MD/Amber99sb-ildn (black circles)].

unfolded random coil regions of multiple proteins coalesce into a proto-filaments[10–13], and (ii) an ordered transition state composed of interactions between hairpin motifs of multiple monomers that convert to the fibril structure[14–17]. There is indirect evidence supporting both mechanisms[2,18]. Ion mobility-mass spectrometry suggests that some peptides can form fibrils from monomeric random coils[10]. NMR diffusion experiments[11] found reversible random-coil-to-beta-sheet transitions in early stages of $A\beta_{12-28}$ amyloid formation, consistent with a proposed model of coil-to-beta-sheet transition for $A\beta_{42}$ dimers[12]. On the other hand, anti-parallel to parallel beta-sheet transitions have been proposed for both $A\beta_{40}$ and $A\beta_{42}$[17] fibril formation, and monomeric hairpins have been identified under different conditions[19], consistent with predicted hairpin propensity in this region[20]. Tau monomers display alternate conformational forms that differ in their ability to seed fibrils[21]. We previously studied a 17-residue segment (295-311) of the tau repeat domain containing the PHF6 aggregation-prone hexapeptide 'VQIVYK' motif[22–25]. We subsequently found hairpin conformations within the structural ensemble using molecular dynamics (MD) simulations and cross-linking mass spectrometry (XL-MS)[26], consistent with NMR studies showing 19–24 percent beta content in this region of the monomer[25,27,28]. Computational approaches could, in principle, provide the necessary structural detail missing from experimental methods to characterize fibril nucleation. For example, MD simulations using a coarse-grained protein model of $A\beta$ show that fibril-compatible structures are sampled as structural excitations within the monomer and dimer ensemble[29,30]. However, computational approaches face challenges with sampling the exponentially large number of conformations accessible to protein complexes, as well as limited experimental validation of proposed nucleation mechanisms.

Therefore, which structural motifs within the disordered ensemble are the building blocks of fibril nucleation, and the molecular rearrangements that constitute the transition mechanism, remain poorly characterized.

Here, we address these challenges using a pipeline integrating molecular simulations with experimental data to characterize the atomistic structure of the transition state and the mechanism by which it can nucleate the fibril. We first correlate the percentage of different structural motifs within the MD-generated monomer ensemble with experimental fibril nucleation kinetics across a tau protein fragment mutant library. Next, we overcome the conformational sampling challenge by using the motifs predictive of fibril nucleation rates as constituent monomers to discover transition-state oligomer complexes via fibril conversion simulations. Using this approach, we find a transition state complex of hairpin trimers. We show that the third hairpin monomer within the trimer complex undergoes a spontaneous orthogonal transformation from an anti-parallel beta-strand to a parallel cross-beta structure in complex with the remaining hairpin dimer, thereby creating a chimeric complex containing a cross-beta face to template subsequent fibril elongation. We quantified the free energy profile of this conversion to establish its thermodynamic favorability. Based on the transition state structure, we designed a disulfide-linked peptide dimer and experimentally showed that it sub-stoichiometrically catalyzes fibril formation of the otherwise non-aggregating wildtype sequence. In contrast, dimers linked in a way that cannot form the transition state structure could not catalyze fibril formation. By using a model system in which fibril nucleation is free from interference by other parts of the protein, these results provide a general mechanism for the core process of fibril nucleation that sets the rate-limiting duration of the lag phase. Finally, MD simulations of

tau k18 repeat domain show that this process is suppressed by long-range interactions that compete with the formation of exposed hairpins necessary to form the fibril transition state, whereas the disease-prone mutants shift the equilibrium in favor of such hairpins.

## Results

### Experimental aggregation speed is correlated with monomer hairpin content across mutants

We carried out extensive molecular dynamics simulations of each one of the 22 sequences in a mutant library of the tau aggregating fragment, which we had previously measured fibril formation kinetics[26] (See Supplementary Fig. 1 and Supplementary Table 1 for sequences). From these simulations, we quantified the monomeric structural ensemble of each mutant peptide. For each mutant, two alternative computational approaches were used in order to test the robustness of the predictions to methodological variation. First, we carried out Replica Exchange Molecular Dynamics (REMD)[31] simulations with the Amber96 force field[32] using GBSA[33] implicit solvent model. As an alternate approach, we carried out multiple equilibrium folding simulations from the extended state starting from different random initial velocities. In the second approach, Amber99sb-ildn[34] force field was chosen to test force field variability (see materials and methods).

The combination of Amber96 force-field with implicit solvent models previously was shown to predict correct folding on a variety of proteins/peptides[35–37] using both equilibrium MD and REMD[38]. To independently validate the force field using REMD and GBSA, we tested the approach on two different de-novo designed small peptides with known structures: a 12-residue long beta-hairpin (1le1.pdb) and a 17 residue alpha helix (2i9m.pdb). Consistent with previous literature studies, the method predicted the correct NMR structures; within 1.2 Å RMSD for the hairpin and 2.4 Å for the longer alpha helix (See Supplementary Fig. 2). These benchmarking data demonstrate that this simulation approach is not biased toward a particular secondary structure for small peptides.

To correlate experimental fibril nucleation rate with MD monomer hairpin content, we computed the fraction of the monomeric ensemble of each mutant peptide in which the peptide is in a hairpin conformation. The hairpin criterion is based on internal pairings of the aggregation-prone hexapeptide residues with the N-terminal side (see Methods). Because of the well-documented importance of PHF6 hexapeptide for tau aggregation[22–25], we set the hairpin criterion to include the entire hexapeptide (other than the highly mobile terminal 311K) being in the beta-sheet form. Hairpins are categorized into two hydrogen-bonding registry classes. "Even" hairpins are those in which even-numbered C-terminal hexapeptide residues 306V, 308I, and 310Y are bound internally to any of N terminal residues, whereas "odd" hairpins are ones in which residues 305S, 307Q, and 309I are hydrogen-bonded to N-terminal residues (see Fig. 1b). Even numbered residues(306V, 308I, and 310Y) are predominantly hydrophobic whereas the odd numbered residues (305S, 307Q, 309I, and 311K) are predominantly polar/charged (Fig. 1b). Alternating polar-hydrophobic residues have long been associated with amyloidogenic sequences[39] and were reported to be disfavored for this reason by evolution[40]. Note that in odd hairpins, the more hydrophobic residues in the even-numbered positions expose their backbone to the solvent. Because amyloid fibrils have intermolecular backbone-to-backbone hydrogen bonding, solvent-exposed backbones of the hydrophobic residues can potentially promote amyloid formation in intermolecular interactions. To gain more insight into this alternating sequence nature and different possible roles of polar and hydrophobic residues, we quantified odd and even hairpin contents separately. For each mutant, the fraction of the structural ensemble that are in the two classes of hairpins or random coil conformation are shown in Fig. 1c. Using these probabilities we ranked the peptide mutants according to random coil content, as well as total hairpin content. These rankings were then compared to the experimental ranking of the mutants according to fibril formation rates as determined by $T_{1/2}$, the time to half-maximum aggregation (see Supplementary Table 1 for experimental ranking and exact hairpin probabilities/rankings).

The correlation plots between MD-generated monomer secondary structure content and fibril kinetics across the mutant library are shown in Fig. 1d, e. The random coil ranking is anti-correlated with the experimental ranking (r = −0.37 and r = −0.47 for REMD/Amber96 and MD/Amber99sb-ildn, respectively), while hairpin ranking positively correlates with the experiment (r = 0.47 and r = 0.76 for REMD/Amber96 and MD/Amber99sb-ildn, respectively). The correlation holds if, instead of rank correlation, we compare hairpin probabilities with the experimental $(1/T_{1/2})$ values (see Supplementary Table 1). The total beta-sheet contents of the hexapeptide segment (requiring all residues to have extended beta sheet, "E" in per-residue secondary structure analysis), rather than the hairpin-based analysis, also shows a consistent positive correlation of 0.44 with the experimental aggregation rates (see Supplementary Fig. 4). These results suggest that the transition state to amyloid formation involves monomeric hairpins rather than disordered motifs.

In addition to the overall correlations between hairpin structure and fibril nucleation kinetics, the MD data provide detailed mechanistic insight into the structural ensemble of individual mutants. For the most common disease-associated mutation P301L[41], the MD trajectories show that the sequence context of the wild-type (WT) proline in the turn region suppresses hairpin formation compared to both P301S and P301L mutants. This is because the rigid proline turn is geometrically compatible with only a subset of hairpin registries. WT peptide folds into two predominant hairpins compatible with the proline turn, both of which contain unfavorable hydrophobic-hydrophilic pairings (Supplementary Fig. 5). For example, in one of the hairpin structures (4.74% of the ensemble), P301 (at *i* + *1* position) plays the role of the downstream partners in the turn region, which is known to be the preferred proline orientation[42,43]. Due to the configurational restraint of proline, there are downstream hydrophobic-to-charged/polar pairings: 306V pairs with 298K, 308I pairs with 296N, and 309V pairs with 295D. This WT fold leaves both 310Y and 311K completely unpaired, and the last residue of the hexapeptide 309V does not form a stable interaction with the corresponding 295D on the N-terminal side. This is a natural consequence of the proline adopting its preferred *i* + *1* role, which makes the N terminal side shorter than the C-terminal side (see Supplementary Fig. 5). When the proline is mutated, we observe higher hairpin content coming from multiple hairpin conformations that contain both a well-defined beta turn and an increased number of downstream hydrophobic pairings. For instance, for P301L, the dominant hairpin registry (7% of the ensemble) is such that hydrophobic pairings of 306V-300V and 309V-297I, as well as polar-polar pairings of 307Q-299H and 310Y-296N stabilize this hairpin conformation (Fig. 4b). Furthermore, the N terminal 295D pairs with C terminal 311K creating a salt bridge sealing the termini (see Supplementary Fig. 5). These results quantitatively support the mechanism in which ordered hairpins nucleate fibril formation and explains the increased fibril forming propensity of the disease-associated P301L/S mutations.

### Hairpin trimers are transition states of the fibril-templating xHAT complex

The correlation between hairpin content and fibril nucleation speed suggests that amyloid formation is somehow fed by monomeric hairpin motifs. To search for a fibril-like cross-beta structural transition, we prepared systems of aggregation-prone P301S dimer and trimer complexes containing combinatorial arrangements of hairpins sampled from the monomer ensemble. First, starting from two hairpins, we prepared side-by-side and face-to-face dimeric systems and performed equilibrium MD simulations in explicit solvent. Even if we bias the

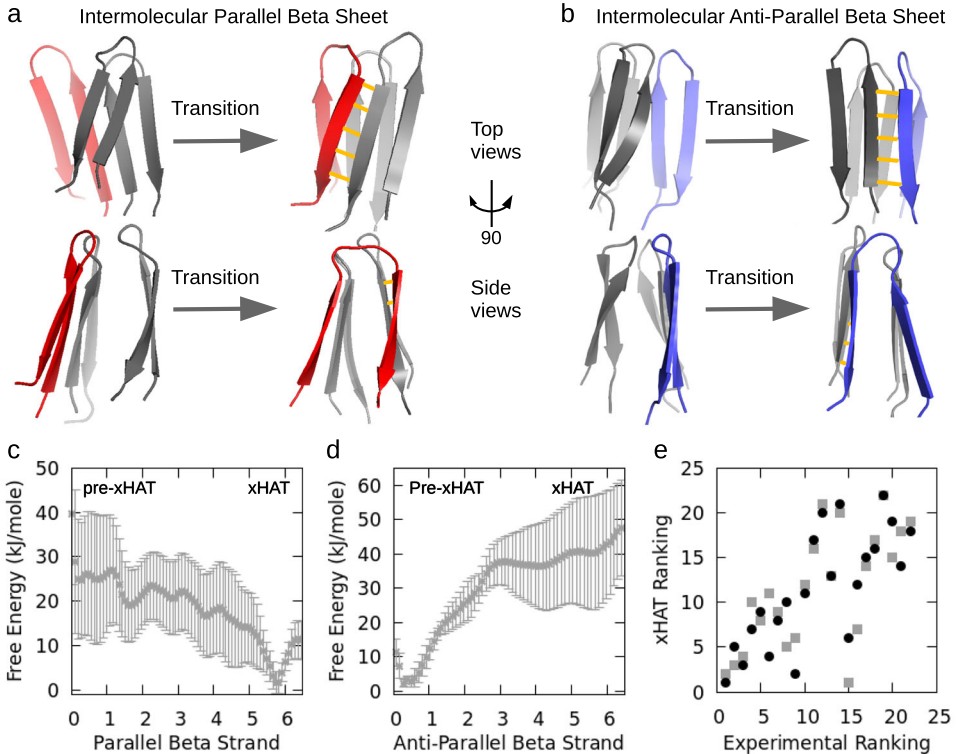

**Fig. 2 | The proposed xHAT model and associated free energy profile. a** The 3rd monomer (in red) is undergoing a transformation from hairpin form into a parallel cross-beta fibril state on the cross-sectional surface of a face-to-face hairpin dimer (in gray). **b** the same transformation of the 3rd hairpin (in blue) for anti-parallel cross-beta transition. **c** Downhill free energy profile for parallel transition, and **d** uphill free energy profile for anti-parallel transition (*n* = 10 independent meta-dynamics trajectories, 5 for each. Data are presented as mean values +/− SD). **e** Correlation between xHAT model and experimental aggregation speeds [r = 0.67 and r = 0.76 respectively by REMD/Amber96 (gray squares) and MD/Amber99sb-ildn (black circles)].

initial geometries toward the fibril state by pairing the two C-terminal hexapeptides and bringing all hydrophobic residues on the same side in the side-by-side orientation or putting all hydrophobic residues into the interface in the face-to-face orientation, the dimer simulations did not produce any fibril-like cross-beta transition (see Supplementary Fig. 6a, b). This suggests that the dimer complex is missing a component for adopting the transition state. Previous experimental studies, such as SEC and XL-MS, have shown that a trimer is the minimal component for unassisted entry and seeding in cells[44]. Additionally, trimeric tau at low concentrations was proposed to be the toxic units to human neuronal cells, rather than monomers and dimers[45]. Therefore, we next tested if trimeric combinations of hairpins from the monomer ensemble could transition to a parallel cross-beta structure. To constrain the conformation space of possible trimers, we considered trimer symmetries compatible with oligomerization. Alternating positioning of hydrophobic residues (304G, 306V, 308I, 310Y) and polar/charged residues (305S, 307Q, 311K) dictates that one side of any hairpin fold is necessarily more hydrophobic than the other side. Because the observed amyloid fibrils bury hydrophobic residues, as expected, topologically three identical hairpins cannot all bury their hydrophobic sides while making an inter-molecular parallel cross-beta transition (see Supplementary Fig. 7). Therefore, one of the monomers should have a shifted registry. This means $i - j$ internal hydrogen bonding must shift to $i - (j \pm 1)$ internal bonding, where $i$ and $j$ are any residues in the n-terminal and hexapeptide sides, respectively. When compared to the original hairpin, a shifted registry flips all hexapeptide side chains and inverts their backbone oxygens and amide hydrogens (see Fig.1b). Therefore, we should either have an odd-odd-even trimeric complex or an even-even-odd complex. As the observed amyloid fibrils have inter-molecular backbone hydrogen bonding, the transition state would be expected to include an odd hairpin with

hydrophobic hexapeptide residues exposing their backbones to the solvent in order to facilitate inter-molecular backbone hydrogen bonding. This leaves the other two hairpins to be even. Therefore, we focused on favorable even-even-odd trimers composed of hairpins enriched from the monomer ensemble.

Equilibrium MD simulations of aggregation-prone P301S mutant even-even-odd hairpin trimers revealed a spontaneous transition of the odd hairpin into a cross-beta structure spanning the interface of a face-to-face interaction between the two even hairpins (Supplementary Fig. 6d). This cross-beta structure is consistent with subsequent templating of additional monomer subunits in the fibril growth phase. When we repeated the same simulations for a trimer with identical hairpins, which would have produced an anti-parallel cross-beta interface, we did not see any transition (Supplementary Fig. 6c). Instead, we observed that each hairpin preserves its anti-parallel beta-sheet structure persistently, as in the case of dimer simulations (see Supplementary Fig.6). This spontaneous finding from non-biased explicit solvent simulations led us further to carry out free energy sampling on both systems (see Materials and Methods). Based on the observed transition in equilibrium MD, we prepared a target trimeric complex with the third hairpin in the cross-beta form which we call the "cross-beta hairpin amyloid trimer" (xHAT). Using the well-tempered metadynamics method (See Methods), we calculated the free energy profile from the hairpin trimer ("pre-xHAT") to the xHAT structure for trimers consisting of either even-even-odd hairpins (converting to parallel beta strand interface with remaining hairpin dimers, see Fig. 2a), or even-even-even hairpins (converting to anti-parallel beta strand interface with remaining hairpin dimer; see Fig. 2b). We obtained a downhill energy path (-25 kJ/mole energy change) for the conversion from the pre-xHAT trimer of even-even-odd hairpins into the xHAT structure (Fig. 2c), explaining our observation of

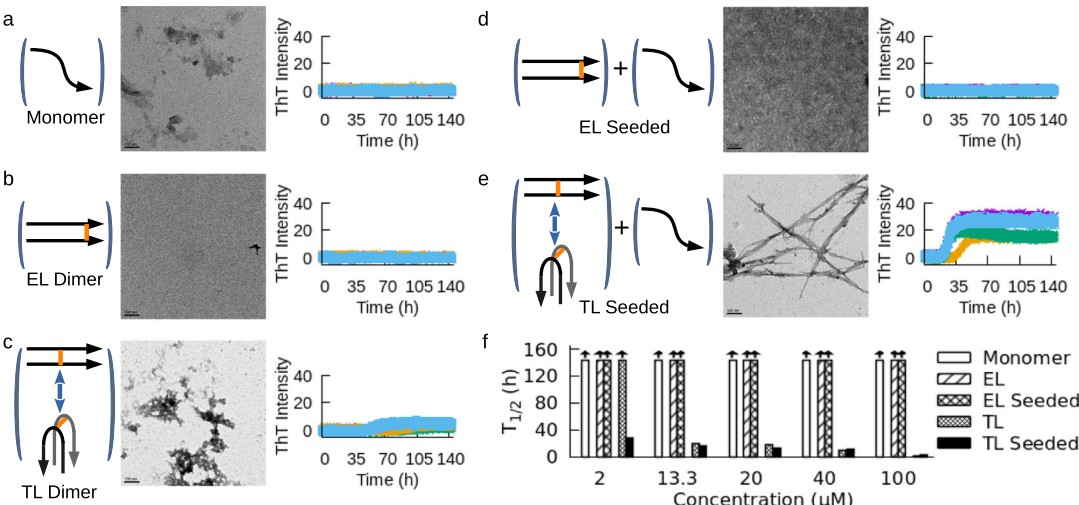

**Fig. 3 | Experimentally testing the xHAT model using engineered dimer catalysts. a** TEM image, and ThT fluorescence intensity of WT monomer showing no fibril formation at 100 µM. End-linked (EL) dimer (**b**) and turn-linked (TL) dimer (**c**) do not form fibril at 2 µM. Representative conformations consistent with dimer constructs are schematized in parentheses. **d** When the non-aggregating WT monomer (200 µM) was seeded by 2 µM EL-dimer, no fibrils were observed. **e** WT monomer (200 µM) was seeded by 2 µM TL-dimer, fibrils are formed within 35 h.

**f** Concentration dependency of the constructs. Monomer, EL and TL dimer constructs by themselves, and monomer seeded by EL-dimer do not form fibrils even at high concentration within 144 h. 2 µM TL dimer is sufficient to seed monomer aggregation (1:100 stoichiometry), whereas 13.3 µM TL dimer is sufficient to seed self-aggregation. All experiments were replicated four times. Quadruplicate ThT experiments are shown in different colors; green, cyan, purple, and brown.

spontaneous hairpin-to-cross-beta transition in the µs time-scale equilibrium simulations. However, for the even-even-even pre-xHAT complex which can in principle form an xHAT complex with an antiparallel interface between the cross-beta structure and the remaining two hairpins (Fig. 2b), an uphill energy path (40 kJ/mole barrier) was computed (Fig. 2d), again explaining why we do not see such conversion in equilibrium MD simulations. The xHAT mechanism suggests a more accurate prediction of fibril nucleation using knowledge of populations of different hairpin registries rather than total hairpin content. Therefore, we calculated an xHAT score (even-even-odd complex explained in the previous paragraph): $S_{xHAT} = f_{odd}f_{even}^2$, where $f_{even}$ and $f_{odd}$ is the fraction of even and odd hairpins in the mutant monomer ensemble, respectively. Using the xHAT score rather than total hairpin content improves the correlation with fibril kinetics (Fig. 2e).

Although it is not possible to experimentally resolve the transition and xHAT states, the xHAT model suggests a structure-based strategy for experimental testing. Namely, a dimeric peptide mimicking two of the three hairpins in the pre-xHAT transition state structure should be able to serve as a catalyst for fibril nucleation. We first show that, consistent with expectation, the WT peptide monomer does not form fibril at 200 µM concentration within the time window of 144 h, as evidenced by both Thioflavin T (ThT) fluorescence and transmission electron microscopy (TEM; Fig. 3a). We next designed two different peptide dimer constructs, which were engineered via disulfide bonds at cysteine mutations at different residues on the WT peptide. In the first construct, which we call the "end-linked" (EL) dimer construct, the C-terminal ends of two monomers are linked allowing formation of a parallel and in-register linear dimer (See schematic in Fig. 3b). In the second "turn-linked" (TL) dimer, a disulfide bond connects the middle of the two monomers, allowing the two linked monomers to be in the parallel orientation as well as the anti-parallel orientation of the face-to-face hairpins consistent with the xHAT structure (See schematic in Fig. 3c). The short distance constraint provided by disulfide bonds (5−7 Å, $C_\alpha − C_\alpha$) restricts the conformational space so that only the TL dimer should be able to seed fibril formation of the monomer via the xHAT mechanism. At 2 µM dimer concentration, none of the dimeric constructs form fibrils on their own (Fig. 3b, c). However, when we

seeded the non-aggregating WT monomer peptide at 200 µM with each of these dimers, only the monomer that is seeded by the TL dimer formed amyloid fibrils (Fig. 3d, e). We repeated these experiments for dimer concentrations up to 100 µM, and the EL dimer was not able to seed at any concentration (Fig. 3f and Supplementary Fig. 8). At 13.3 µM, TL dimer was able to form fibrils on its own, which is consistent with the xHAT mechanism because two TL dimers can in principle combine to contain a tetramer of hairpins consisting of a trimeric pre-xHAT sub-system. Fibril formation requires ThT signal change as well as observed fibrils in TEM. These results show that fibril nucleation must satisfy geometric constraints beyond non-specific association. Furthermore, the transition state is not the simple linear form consisting of parallel intermolecular beta strands, which would be consistent with catalysis by the EL dimer.

## Dynamics of the entire tau repeat domain puts the xHAT mechanism in context

To place the xHAT mechanism within the context of protein-scale conformational heterogeneity, we performed MD simulations on the entire k18 domain of tau protein, using WT and two disease associated mutants (P301L and P301S). The k18 domain (244 to 372[46]) consists of 4 homologous repeat domains separated by PGGG motifs (Fig. 4a and Supplementary Fig. 1). The first three repeat domains, containing the [275]VQIINK[280], [306]VQIVYK[311], and [337]VEVKSE[342] hexapeptides, are known as PHF6*, PHF6, and ("Module-B"[47]) domains, respectively. Note that the 17-residue peptide fragment that forms the xHAT nucleation mechanism is situated at the H2 region within k18 at the junction between R2 and R3 (Fig. 4a). All three hexapeptides, but especially PHF6* and PHF6, are known to be crucial for tau aggregation[22–25]. As REMD would be computationally limited for such a long sequence, we performed independent folding simulations, totaling 2 µs (2000 folds, each with 1 ns) for each mutant, starting from linear structures. We then quantified local hairpin probabilities surrounding each PGGG motif (from H1 to H4) as well as the pairwise beta sheet formations among 4 different hexapeptides [1 = [275]VQIINK[280](PHF6*), 2 = [306]VQIVYK[311](PHF6), 3 = [337]VEVKSE[342](Module-B), and 4 = [369]KKIETH[374]] (Fig. 4a, b). We found pairings of these segments in a variety of combinations, producing transient structures containing up to 15% beta structure. For WT k18,

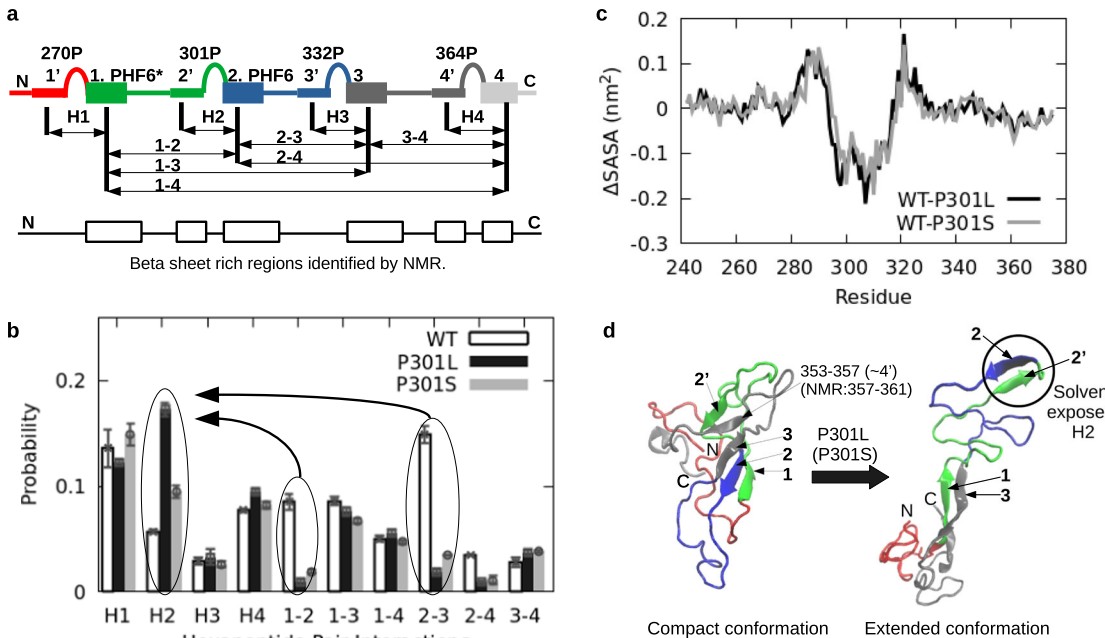

**Fig. 4 | Local hairpin folds and long range inter-hexapeptide interactions in k18. a** Cartoon representation of four hexapeptides (1,2,3, and 4) after each PGGG, and their n-terminal flanking sites (1', 2', 3', and 4') in four repeat k18. R1 is in red, R2 is in green, R3 is in blue, and R4 is in gray. Below, beta-sheet propensities as determined by NMR, boxes correspond to regions of at least 4 contiguous residues of negative secondary chemical shifts averaged for C-alpha and C' (data from ref. 25). **b** Probability of local hairpins around each PGGG motifs (H1 to H4) and all possible inter-hexapeptide interactions; $n = 2000$ independent folding simulations. Data are presented as mean values +/− SD. **c** SASA values; first plot is for WT-P301L and the second one is for WT-P301S (+y values mean WT is more solvent exposed while -y values mean mutant is more exposed for a given residue). **d** Representative MD snapshots of WT and mutant k18, where a solvent exposed local hairpin (H2) emerges under both P301L (shown) and P301S mutations. Hexapeptides start at the beginning of each repeat domain.

the most dominant beta formation was due to pairing between the second (PHF6) and third (Module-B) hexapeptides (15% probability). Averaged secondary NMR chemical shifts indicate that soluble k18 has beta sheet propensities in three main regions, each containing one of the hexapeptides[25]. The beta propensity of these three regions is also evident in k32[27]. Our results show that these residual beta formations are mainly due to long-range inter-hexapeptide interactions of 1-2, 1-3, and 2-3 (Fig. 4b). MD data also suggests relatively less involvement of the last hexapepide (1-4, 2-4, and 3-4 beta formations) compared to mutual pairings of the previous three (Fig. 4b). In addition to long-range hexapeptide interactions forming beta sheets, we also observed orthogonal folds in which hexapeptides pair with their n-terminal flanking sites to form local hairpins. In WT k18, H1 and H4 local hairpins are favored over H2 and H3 hairpins (Fig. 4b)). This is because hexapeptides 2 and 3 form more favorable long-range interactions that compete with local hairpin formation. It has been noted that the N terminal flanking segment of the PGGG motifs 295D-299K (2' in Fig. 4a) and 357L-361T (4' in Fig. 4a), possess beta sheet propensities, but interestingly not the corresponding PGGG motif in R3 (325L-329H, 3' in Fig. 4a)[25]. This is consistent with our H2 (2-2') and H4 (4-4') local hairpin predictions. Although missing NMR beta sheet propensity of 3' is consistent with the very small probability of H3 local hairpins, it directly points to other non-local interactions responsible for the beta sheet propensity of the third hexapepide (3); we predict this to be the 2-3 interactions. Interestingly, the most dominant WT beta formation (2-3) almost completely disappears for both P301L and P310S mutants in favor of more local H2 hairpins (Fig. 4b). From the solvent-accessible surface area (SASA) per residue (Fig. 4c), H2 hairpins are much more solvent-exposed in these mutants compared to WT. That is, mutating P301 not only increases local H2 hairpin propensity but also increases hairpin exposure to solvent. Representative MD snapshots shown in Fig. 4d reflect these changes in solvent exposure due to the increased H2 hairpin content in the P301 mutants; the exposed H2 hairpin can

subsequently initiate inter-molecular fibril nucleation via the xHAT mechanism.

The xHAT mechanism can explain structural commonalities between patient-derived tau fibrils across multiple tauopathies. The tau isoforms containing the 295-311 segment modeled here all share a turn structure with heterosteric zipping of the n-terminal and c-terminal sidechains, similar to the third monomer of the xHAT complex. A recent structure-based classification of tauopathies associates fourteen different diseases to eight different tau fibril structures, and the heterosteric zipper in the 295-311 segment always appears if the tau isoform contains both R2 and R3 repeats (see Fig. 3 of ref. 48). Three out of the eight fibril structures do not exhibit the 295-311 turn structure as they are all missing the sequence segment. Structures of all five 4R tauopathies show very good overlap when 295-311 turns are superimposed with the final cross-beta monomer in the xHAT model (Supplementary Fig. 9). Additionally, filaments of FTDP-17 associated P301S tau mutant from transgenic mouse brain have recently recently been resolved[49] (the P301L fibril structure is not yet resolved). The structure extends from K274 to H329 and has a central turn around the 295-311 region with heterosteric zipping of the hydrophobic residues. These data suggest that the monomeric hairpin fold in 295-311 tau segment is the preferred location for fibril nucleation under disease conditions if the isoform contains this segment. For isoforms that do not contain this segment, the xHAT mechanism can nucleate at other vulnerable sites with high hairpin propensity. For example, the Alzheimer's disease filament (6hre.pdb) has a central turn and heterosteric hydrophobic zipper around K343-K353 segment. Our model suggests that this fibril structure may be seeded by a monomeric hairpin in this segment in which there is an almost perfect alternation of polar/charged and hydrophobic residues. A hairpin with a turn containing K347-D348-R349 charged residues can place the remaining odd numbered residues (all polar/charged: K343, D345, Q351, and K353) on one face of the hairpin while leaving the even-

numbered residues (all hydrophobic except S352: L344, F346, V350, and I354) on the other side. Such hairpins that are polar/charged on one side while hydrophobic on the other side are biophysically suitable for the pre-xHAT to xHAT transition, consistent with the mutants of the 295-311 peptide that we observed to be aggregation prone.

In addition to WT NMR data, our results also explain two key NMR observations. First, the k18 P301L mutant has an identical spectra with WT, having the same beta propensity regions with no increases in the observed beta propensities[50]. Our findings explain this observation: the loss of WT 1-2 and 2-3 inter-hexapeptide beta formations are compensated by the increase in local H2 (2-2') hairpins in P301L mutant. The H2, 2-3, and their total probabilities are determined to be 0.06, 0.15, 0.21 for WT, and 0.17, 0.02, 0.19 for P301L, showing conserved *total* beta formation for hexapeptide 2, although the mutation induces a significant shift from inter-hexapeptide beta formation to a local hairpin fold around the mutation site (Fig. 4b). Secondly, NMR data suggests that the PGGG motif has a turn structure in WT but in P301L this region converts into a more extended beta sheet which also includes the previous three residues of K298-V300. This can also be explained by our results: the increase in H2 hairpins bring all four residues of K298-301L segment into an extended beta sheet. Finally, NMR also shows that k18 (R2R3) has the same NMR pattern as k19 (R1R3) despite missing the R2 domain containing hexapeptide 2[25]. As shown in (Fig. 4b), we find that the 1-3, 1-4, and 3-4 interactions are not sensitive to the involvement of hexapeptide 2 in overall pairwise interactions, consistent with this NMR observation.

In all NMR spectra, the hexapeptide sides exhibit more dominant beta propensities compared to their n-terminal flanking sides. This intriguing observation is explained by inter-hexapeptide beta sheet formations leaving their n-terminal flanking segments (1', 2', 3', and 4') as disordered/helical, thereby reducing their beta propensities. Only local hairpins contribute to the observed beta sheet propensities in 1', 2', 3', and 4' segments whereas both local hairpin conformations and inter-hexapeptide beta conformations contribute to the propensities of 1, 2, 3, and 4. Such unequal compensatory contributions coming from these two predicted orthogonal conformations can give rise to the observed asymmetry around the PGGG motifs in NMR spectra.

## Discussion

In contrast to simple crystallization, previous studies suggest that amyloid fibril formation should be described by a multi-step nucleation mechanism in which oligomeric intermediates play crucial roles[51]. In particular, a widely accepted model called "Nucleated-Conformational-Conversion (NCC)" formulated by two experimental[52,53] and a later theoretical study[54], suggests that critical nuclei form by key structural conversions at some oligomeric state. More recent studies also support oligomer-driven fibril formation[55–58]. In the NCC model, the nature of the rate-limiting step remains unresolved. Although it is known that an oligomer plays a central transition-state role, the atomistic structure of such an oligomer remained elusive. Furthermore, it was unclear if the rate-determining step is from monomer to oligomer, or from oligomer to fibril. Here, our study proposes a well-defined trimeric hairpin complex as the key oligomer structure, and predicts the rate-limiting step to be the formation of this oligomer from monomeric hairpin conformations.

By correlating experimental fibril kinetics with structural features from extensive MD simulations across a library of tau peptides, we find that beta hairpin content within the intrinsically disordered monomer ensemble–and not coil content–is positively correlated with fibril-nucleation rate. This work therefore quantitatively validates the mechanism whereby the parallel cross-beta fibril structure nucleates from specific anti-parallel hairpins, challenging the previous picture of an unstructured transition state formed from disordered loops[59,60]. This mechanism has been suggested by previous work. For example, sequence features such as charged residues within the hydrophobic

faces of natural β-rich proteins have been proposed as evolutionary design principles to protect against inter-molecular beta sheet interactions that may lead to aggregation[61]. Consistent with this picture, fluorescence correlation spectroscopy showed cooperative unfolding of monomeric Aβ42[62]. A beta-hairpin motif was found in α-synuclein, and sequestration of this hairpin was shown to inhibit fibril formation[14]. Modification of the β-turn region of the hairpin altered the fibril morphology[63], and a sequence region overlapping this hairpin motif was found to control α-synuclein aggregation fate[64]. Beta-hairpin mediated fibril nucleation was also suggested for Huntington fibrils[65,66], as well as for the monomeric ensemble of human islet amyloid polypeptide (IAPP) but not for the non-amyloidogenic rat IAPP which is only 6 residue different[67,68]. However, previous work fell short of systematically correlating the structural motifs such as hairpin content with aggregation tendency or speed. In addition, no molecular mechanism for the nucleation process had been proposed and tested.

Using the transient hairpins collected from MD simulations as building blocks, we show that the transition state is a trimer of hairpins, rather than a dimer, consistent with literature data reporting tau trimers are key to amyloid formation[44,45]. If one of the hairpins in the trimer complex is registry-shifted with respect to the other two, we demonstrate using both equilibrium MD and meta-dynamics MD sampling that the shifted hairpin can spontaneously transition to a parallel cross-beta (amyloid) fold stabilized by the remaining face-to-face hairpin dimer. Simulations further indicate that the cross-beta transition is energetically unfavorable if all three hairpins in the trimer complex have the same registry. This might explain why structures like beta barrels, which are formed from identical hairpins[69,70], prefer anti-parallel side-by-side assembly rather than fibril-like cross-beta structure. On the other hand, if the transition state involving a registry-shifted hairpin is kinetically accessible, then the system will convert to the more thermodynamically favorable parallel-cross beta structure. It is worth noting that, for well-folded proteins, folding of a given local sequence into two alternative hairpin registries is extremely unlikely. However, local sequences in disordered proteins like tau, amyloid-beta, and a-synuclein, can potentially fold into hairpins with two or more different registries. Therefore, the unique secondary structure folds of non-aggregation-prone proteins seems to be a kinetic constraint that protects them from their thermodynamic fate, which is the ordered fibril state[3].

We experimentally tested the xHAT model by designing turn-linked (TL) disulfide peptide dimers that can act as a catalyst for xHAT-based fibril nucleation of WT tau peptides that do not form fibrils on their own. As a control, we designed an end-linked (EL) dimer peptide that was crosslinked in a way that could form proto-fibrils but not the xHAT structure. Both ThT fluorescence assays and electron microscopy images revealed that only the xHAT-compatible TL dimer construct could catalyze amyloid formation of WT peptide, strongly supporting the xHAT model. Importantly, the EL-dimer compatible with a linear proto-fibril is unable to catalyze fibril formation of WT tau peptide at any concentration. This is consistent with studies of α-synuclein in which both disulfide linkage of the two so-called master-controller segments (residues 36-42) via V40C mutation[64], and dimerization via two Y39 di-tyrosine bonds[71] inhibit aggregation. It is apparent that both of these links (39-39 and 40-40) are from the middle of the aggregation-prone master controller (P1, 36-42) which are not compatible with a face-to-face hairpin dimer requiring around 1 nm separation, but perfectly compatible with an extended beta sheet dimer containing both disulfide and di-tyrosine inter-molecular links.

Folding simulations on the entire tau repeat domain (k18) and two of its aggregation-prone mutants (P301L and P301S) not only produced key interactions in WT and mutants, but also brought independent support for hairpin based xHAT model. We found that hexapeptides at the beginning of each repeat domain undergo mutual long-range beta formations that keep WT k18 compact compared to

the mutants. In particular, beta sheet formation between the second hexapeptide (PHF6) and the third one (module B) is the most dominant interaction in WT. Consistent with our predictions, recent AFM[72] and chemical cross-linking[26] experiments report WT to be more compact compared to aggregation prone mutants. The AFM study suggests up to 3 beta-sheet (unfolding) events captured per molecule in compact k18 conformations. There is no correlation in the force extension curves, pointing to non-specific conformers[72]. Our data explains such non-correlated capture events as arising from different beta-sheet formations in different inter-hexapepetide interactions. We further observed that both P301L and P301S mutants disturb the most dominant inter-hexapeptide interaction (PHF6-ModuleB) in favor of more local hairpins around the second PGGG motif. Mutants not only give rise to more local hairpins but also make them more solvent exposed. These findings are in agreement with NMR data on both soluble WT k18 and its P301L mutant. These results on the entire k18 domain support our findings of the peptide model system indicating that amyloid formation is nucleated by hairpin oligomers.

The xHAT model has implications for explaining fibril heterogeneity and targeting fibril formation, warranting future study. First, the different fibril morphologies observed for the protein under different conditions (such as cofactors, post-translational modifications, and mutations) can be explained by xHAT seeding from different local hairpins that are differentially affected by the conditions. This picture is consistent with recent evidence of tau mutations that promote amyloid formation while stabilizing local structure[73]. This is especially important for tau where different neurodegenerative diseases are associated with different fibril morphologies. Second, the pre-XHAT transition state complex suggests a potential target for kinetic modulation of fibril formation via inhibition of either face-to-face dimer formation or cross-beta transition of the third hairpin.

## Methods
### Simulation details

In this study, four different sets of molecular dynamics (MD) simulations were performed, each with a different methodology. To obtain reliable conformational ensembles, we employed two independent methods; (1) enhanced sampling with Replica Exchange MD (REMD)[31] and (2) equilibrium multiple fast folding simulations starting from linear structures. In both of these folding simulations, we used an implicit solvent model to facilitate the sampling of highly heterogeneous structural ensembles. The third simulations were carried out in explicit solvent to search for possible fibril-like cross beta transitions in dimeric and trimeric hairpin spaces. Finally, as the fourth one, well tempered metadynamics[74] were employed to get free energy profiles, again using explicit solvent molecules.

In REMD folding simulations, we utilized a widely used implicit solvent model, Generalized Born with the hydrophobic solvent accessible Surface Area GBSA[33]. Amber96 force field[32] was chosen as it was previously shown to predict correct structures in folding simulations when combined with implicit solvent models[35–37]. We started from fully extended linear structures and generated replicas at 6 different temperatures (290.00 K, 310.20 K, 331.42 K, 353.68 K, 377.05 K, 401.59 K) while attempting exchanges at every 2 ps. At each temperature, 1 μs long simulation was done with 1 fs time steps, producing a total of 6 μs long total sampling per REMD run. This REMD set-up showed good energy overlaps with exchange probabilities around 0.35–0.40 (see Supplementary Fig. 3). We performed 10 independent REMD runs starting from different random velocities, obtaining a total of 60 μs total simulation data for each peptide [10×(6 × 1μs)]. Overall, we generated a total of 1320 μs (=1.32 milliseconds) REMD sampling over 22 different mutant libraries. In final analyses, we collected statistics from the lowest 3 temperatures (290.00 K, 310.20 K, 331.42 K). The methodology was first tested on two peptides with known

structures. Starting from linear structures and employing the exact details above, we show that REMD/GBSA/Amber96 method is able to produce correct NMR structures of a 12-residue long beta-hairpin (1le1.pdb) and a 17-residue long alpha helix (2i9m.pdb) within 1.2 Å and 2.4 Å RMSD errors respectively (See Supplementary Fig. 2) To independently validate REMD results, a second set of alternative folding simulations were performed. Here, we again used GBSA implicit solvent model but with a different force field, AMBER99sb-ildn[34]. Rather than REMD, we now carried out 5000 equilibrium folding simulations per mutant at T = 300 K, with no bias/perturbation. At each cycle, random initial velocities were assigned and the initial linear structure is let to fold within 1 ns long simulation. Because the mutant peptides are short (17–19 residues), such multiple-short simulations were able to produce a different fold topology at each cycle. The ensemble of final structures, that is 5000 per mutant, was used in post-simulation analyses.

In explicit solvent equilibrium simulations, a dodecahedron box with periodic boundary conditions was constructed for each peptide. Simulation boxes were solvated using SPCE[75] water molecules, with a minimum distance of 1.5 nm between solute and box edge. After an initial steepest descent energy minimization, 10 ns NVT, 10 ns NPT with Berendsen[76] barostat, and 10 ns NPT with Parrinello-Rahman[77] barostat were performed for thermal equilibration. 2 fs time steps with Particle Mesh Ewald (PME)[78] summation for long-range electrostatics were employed. NPT ensemble, with Nose-Hoover[79] thermostat and Parrinello-Rahman[77] barostat were used for all production-level simulations. The minimal number of neutralizing ions (either Na or Cl) were added to each box to ensure charge neutrality. AMBER99sb-ildn force-field[34] was utilized due to its accuracy in describing protein structure[80].

For free energy samplings, we performed well-tempered metadynamics[74] and introduced perturbations using the *parabetarmsd* and *antibetarmsd* collective variables (CV) as implemented in plumed[81] package. In parallel cross beta transition, the two C-terminal hexapeptides (one in the 3rd hairpin and the other on the opposite side) are included in *parabetarmsd* CV. In anti-parallel transitions, the same hexapepdite in the 3rd hairpin was now paired with N-terminal residues (295D to 300V) on the opposite side and included in *antibetarmsd* CV. Using preliminary runs, we tuned the well-tempered metadynamics parameters to be 1.2 kJ/mol, 0.3 kJ/mol, and 15, for gaussian height, CV gaussian width, and the bias factor, respectively. Time-dependent gaussian hill potentials were included into MD at every 2.5 ps. We generated 10 independent metadynamics runs (5 for each), getting a total of over 6 μs sampling for cross-beta transitions. Metadynamics simulations were performed using explicit solvent molecules, with the same set-ups and details as explained above. All molecular dynamics simulations (MD) were performed using GROMACS 5.0.4[82] software patched with PLUMED-2.3.7[81] on UTSW's biohpc cluster.

In post-simulation analyses, hairpins were quantified using the following two criteria; (1) at least five residues (from 306V to 310Y) of the C-terminal PHF6 hexapeptide must be in the hairpin, and (2) hexapeptide residues internally bound to N terminal residues must form proper hydrogen bonds with a heavy atom distance cut off of 3.5 Å. The terminal residues (295D and 311K) were not taken into consideration as they are exhibiting mobile dynamics. A hairpin is called "even" if even-numbered hexapeptide residues (306V, 308I, and 310Y) make internal hydrogen bonds, and "odd" if odd-numbered residues (305S, 307Q, and 309I) are hydrogen-bonded to N-terminal residues (see Fig. 1b) In quantifying random coils, we used STRIDE[83] as implemented in VMD[84] package and calculated per residue secondary structure assignments. Number of hexapeptide residues having random coil assignment ("C" in stride notation) were summed over all frames and then divided by the proper normalization constant, 6*(number of frames), to get the probability of random coil conformations for each mutant. Note that for both hairpin and random

coil quantifications, the PHF6 hexapeptide was especially taken as the reference point. That is, hairpins are those having hexapeptide in antiparallel beta-sheet with N-terminal residues, and random coils are those having one or more hexapeptide residues with random coil secondary structure assignments as defined by "C" in stride. Therefore, our metrices for hairpin and random coil conformations are mutually orthogonal, such that a given fold cannot be counted in both.

### Experimental details

Three different disulfide-linked dimers of WT peptide were designed as explained in the main text. The miss-linked (ML) dimer is (DNIKHVPGC GSVQIVYK)–(DNIKHVPGGGSVQIVYKC), the end-linked (EL) dimer is (DNIKHVPGGGSVQIVYKC)–(DNIKHVPGGGSVQIVYKGC), and the turn-linked (TL) dimer has the sequence of (DNIKHVPGC GSVQIVYK)– (DNIKHVPGC GSVQIVYK). The WT monomer peptide is 295DNIKHVPGGGSVQIVYK311, of tau. All peptides were synthesized by Genscript with N-terminal acetylation and C-terminal amidation modifications and purified to greater than 95 percent purity.

**ThT fluorescence aggregation assays.** Lyophilized peptides were dissolved in 200 uL trifluoroacetic acid TFA(Pierce) and incubated at room temperature (RT) for 16 h in a chemical fume hood. The peptide solution was dried under a stream of nitrogen or CO2 gas and then immediately placed under a lyophilizer to remove any residual volatile solvents. The peptide residue was resuspended in water and diluted to the desired concentration with the addition of $10 \times$ PBS to adjust the peptide to $1 \times$ PBS buffered reaction conditions. ThT was added to the samples at a final concentration of $25\,\mu M$. $45\,\mu L$ of peptide mix was added in quadruplicate in a 384-well clear bottom plate. All conditions were done in quadruplicates at 37C. For seeded reactions, ThT kinetic scans were run every 30 min with shaking at 800 rpm 10 s prior to each data acquisition on a Tecan Spark plate reader at 446 nm Ex (5 nm bandwidth) and 482 nm Em (5 nm bandwidth). For monomer aggregation reactions, ThT kinetic scans were run with constant shaking at 800 rpm on a FLUOStarOmega at 448 nm Ex (10 nm bandwidth) and 482 nm EM (10 nm bandwidth). Values of the blank wells containing buffer and ThT were subtracted from the values of the experimental groups. $T_{1/2}$ fits for the ThT fluorescence aggregation data were calculated in GraphPad Prism 9.4.1 using the linear regression sigmoidal fit.

**Transmission electron microscopy.** An aliquot of $5\,\mu L$ sample was placed onto glow-discharged Formvar/Carbon Square 200-mesh copper grids for 1 min, washed with distilled water for 1 min, and then negatively stained with $2\,\mu L$ 2 percent uranyl acetate for 1 min. Images, shown with 100 nm scale, were acquired on a Tecnai G212spirit transmission electron microscope (FEI, Hillsboro, OR), serial number: D1067, equipped with a LaB6source at 120 kV using a Gatan ultrascan CCD camera.

### Reporting summary

Further information on research design is available in the Nature Portfolio Reporting Summary linked to this article.

## Data availability

The MD trajectory data generated in this study, along with analysis scripts, as well as experimental ThT values have been deposited in the Zenodo database under accession code 10.5281/zenodo.10456884 [https://zenodo.org/records/10456884]. The MD-derived data generated in this study are provided in the Supplementary Information/ Source Data file. Source data are provided with this paper.

## Code availability

All MD simulations and analyses were performed using Gromacs-5.0.4 (available at http://www.gromacs.org), Plumed-2.3.7 (available at https://www.plumed.org/), and VMD-1.9.4 (available https://www.ks.uiuc.edu/Research/vmd/).

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

## Acknowledgements

We thank Kimberly Reynolds and Marc Diamond for feedback that improved the manuscript. This research was supported in part by the computational resources provided by the BioHPC supercomputing facility located in the Lyda Hill Department of Bioinformatics, UT Southwestern Medical Center, TX. This research was supported by NIH MIRA R35GM150897-01 (L.S and M.M.L), NIH R01AG076459-01 (L.A.J.) and NIH F31NS127513-01 (S.B).

## Author contributions

L.S. and M.M.L. conceived the work, L.S. performed simulations and computational analysis, S.B. performed experiments, L.J. and M.M.L. supervised the work.

## Competing interests

The authors declare no competing interests.
