## [Peer Review File · Nature Communications]

Reviewers' Comments:

Reviewer #1:

Remarks to the Author:

This manuscript addresses two important issues associated with tau protein fibrillation in neurodegenerative diseases: how the process is initiated through nucleation motif 306VQIVYK311 and how disease-causing substitutions at P301 promote aggregation propensity. These problems are approached through a novel experimental pipeline disclosed in the manuscript involving computational and biochemical modeling focused primarily on a 17-mer peptide spanning tau residues 295-311 but also leveraging tau truncation constructs. It also assumes a homogeneous nucleation mechanism. By combining large-scale molecular simulations with previously published experimental data, authors identify a cross-beta hairpin amyloid trimer (xHAT) transition state, and then investigate the trimer through an elegant experimental system involving disulfide cross-linked peptides. They also provide evidence that disease-associated mutations at P301 relieve gatekeeper function in this region. Overall the experiments are well performed, but significance to the field can be strengthened. Specific concerns are listed below.

Specific points

1. Findings are not described in the context of human disease

a. Tau filament polymorphism. Despite a wealth of 3D structure data on disease-derived tau aggregates, authors do not relate their models to any disease-related polymorph. The significance of the findings derived from peptide models would be strengthened if they were to do so. For example, this information may identify under which disease conditions the proposed nucleation scheme could be operating. It would also provide a test of the hairpin probabilities calculated in the manuscript.

Interestingly, certain polymorphs composed of 4-repeat tau such as the Progressive Supranuclear Palsy (PSP) and Corticobasal Degeneration families do form heterosteric zippers in the 295-311 region similar to those investigated in the manuscript. The PSP family in particular contains additional examples of heterosteric zippers proposed in Fig. S1. On the other hand, AD polymorphs do not form these structures (does this mean the proposed mechanism is not operating in AD?). Failure to place its findings into the context of disease is a major weakness of this manuscript.

b. Nomenclature. 3D structures of filamentous tau aggregates from multiple diseases are now available, revealing architectures consisting of proto-filaments and filaments. Fibrils have been observed but only in vitro. In contrast, the manuscript uses the term "fibril" in a different sense and creates the term "protofibril" which has no established meaning. Authors should harmonize their nomenclature with the tau structure field so that readers can better understand their findings.

2. Experimental conditions need clarification

a. Authors work with 200 micromolar peptide, which is far above physiological tau concentrations. Authors should justify the conditions

b. Two constructs termed k18 and k19 are employed in the manuscript but not explicitly defined or referenced. Their nature is particularly confounding because deletion mutants named K18 and K19 have been used for many years in the tau field. Both K18 and K19 lack the C-terminal portion of the 4th microtubule binding repeat (as defined by sequence alignment), making them incomplete models of the microtubule binding repeat region. Authors should explicitly define k18 and k19 and explain why these specific constructs were chosen for experimentation.

c. The depiction of parallel and anti-parallel beta sheets in Fig. 2a,b is unclear; an emphasis on them being intermolecular anti-parallel (or parallel) interactions would be appreciated by journal readers.

3. New methodology has not been fully interrogated

a. Experimental verification of the simulation method (Fig. S2) leverages only two peptides of different lengths. The most accurate helix reproducing structure occurred only 33% of the time. It is not clear that novel methodology has been adequately vetted prior to its application to tau peptides.

b. A justification for the xHAT score (lines 187-189) should be provided to ensure that it is not arbitrarily defined to improve fit.

Reviewer #2:

Remarks to the Author:

This manuscript presents important insights into the nucleation of fibril formation based on a combination of detailed computer simulations and cross-linking experiments. The cross-linking experiments are particularly strong evidence that the trimeric structure the authors have identified, referred to as the xHAT model, is the rate-limiting transition state for nucleation. The simulations were key for designing this experiment and help make sense of the data. The results are non-obvious, since a trimer that mimics the final structure of the fibril is insufficient to catalyze fibril formation.

My major concern is with how the data are presented. The story, as told, is non-linear and makes jumps that are hard to follow. I almost gave into the temptation to write the results off and put the paper down at a couple of points. I don't think additional experiments are needed, but some significant restructuring of the manuscript/figures is warranted. Here are some of my concerns.

- I wasn't clear why the authors started with simulations of the monomer, given that there is existing evidence an oligomer is the rate-limiting step and the authors seemed to agree with this conclusion. I think an appropriate hypothesis is that the oligomer is made up of energetically favorable monomer conformations, but neither this hypothesis or any competing hypothesis was ever given to motivate the initial work. Some hypothesis should be given for why the hairpin content of the monomer is predictive of the rate of fibril formation even though the rate-limiting step is a trimer.

- Adding to the confusion, the authors group monomers into different registries without a clear rational/hypothesis for doing so. Figure 1 also refers to the xHAT state, which is not even been defined in the first few sections of the manuscript (i.e. on the monomer and dimer). These data should either be motivated earlier on, or moved to later in the paper to help explain data on the trimer.

- Similarly, there was not a clear rational/hypothesis motivating the work on the dimer. I think the authors meant to show that the dimer is not the rate-limiting step. But I didn't feel like they started to address that until after talking about results on the dimer, leaving me confused. By the end of the discussion of the dimer, I was just confused what the authors were trying to show and what I was supposed to take away from any of the data they presented.

Responses in **bold**, changes in the manuscript indicated with underlining.

Reviewer #1 (Remarks to the Author):

This manuscript addresses two important issues associated with tau protein fibrillation in neurodegenerative diseases: how the process is initiated through nucleation motif 306VQIVYK311 and how disease-causing substitutions at P301 promote aggregation propensity. These problems are approached through a novel experimental pipeline disclosed in the manuscript involving computational and biochemical modeling focused primarily on a 17-mer peptide spanning tau residues 295-311 but also leveraging tau truncation constructs. It also assumes a homogeneous nucleation mechanism. By combining large-scale molecular simulations with previously published experimental data, authors identify a cross-beta hairpin amyloid trimer (xHAT) transition state, and then investigate the trimer through an elegant experimental system involving disulfide cross-linked peptides. They also provide evidence that disease-associated mutations at P301 relieve gatekeeper function in this region. Overall, the experiments are well performed, but significance to the field can be strengthened.

We appreciate the reviewer's assessment of the importance, novelty, and rigor of our work. We thank the reviewer for careful reading and critical comments, which we have endeavored to respond to below and address in the manuscript. We believe that changes to the manuscript stimulated by the reviewer's comments have greatly improved the work, especially in regards to clarity and the significance to the field. We are grateful for the reviewers time and consideration.

Specific points

1. Findings are not described in the context of human disease

a. Tau filament polymorphism. Despite a wealth of 3D structure data on disease-derived tau aggregates, authors do not relate their models to any disease-related polymorph. The significance of the findings derived from peptide models would be strengthened if they were to do so. For example, this information may identify under which disease conditions the proposed nucleation scheme could be operating. It would also provide a test of the hairpin probabilities calculated in the manuscript.

Interestingly, certain polymorphs composed of 4-repeat tau such as the Progressive Supranuclear Palsy (PSP) and Corticobasal Degeneration families do form heterosteric zippers in the 295-311 region similar to those investigated in the manuscript. The PSP family in particular contains additional examples of heterosteric zippers proposed in Fig. S1. On the other hand, AD polymorphs do not form these structures (does this mean the proposed mechanism is not operating in AD?). Failure to place its findings into the context of disease is a major weakness of this manuscript.

The referee points out an important point, which we did not flesh out in the original submission: the relevance of our xHAT model to different tau morphologies observed under different diseases. Although we used the 295-311 peptide fragments as model systems, our hairpin driven model is not restricted to the hairpin formation in this specific tau segment. Our model is consistent with amyloid formation seeded by hairpin folds rather than by unstructured random coils. It suggests a general mechanism whereby in-register parallel cross-beta amyloid fibrils can be initiated by a trimeric complex of monomeric hairpins whose propensity and location within the chain is dictated by protein sequence. Therefore, our results suggest that different fibril morphologies associated with different diseases could be the result of seeding from

different local hairpin folds stabilized by different biochemical conditions (mutations, PTMs, etc.). Although this point is discussed in the last paragraph of the discussion section, the message needed to be reinforced earlier in the manuscript. To emphasize this significance, we have now included an additional sentence at the end of the abstract: “The proposed mechanism implies that different fibril morphologies observed for the same protein under different disease conditions could be due to disease-dependent seeding of hairpins in alternate protein regions.”

As noted by the referee, the heterosteric zipper in tau 295-311 segment is observed in fibrils extracted from patients with PSP family and CD diseases. Based on a recent structure-based classification of tauopathies (Yang Shi et al, Nature, 2021, pg. 359-363; see Fig. 3), all tau fibril structures interestingly do have a turn and heterosteric zipper in 295-311 region if the isoform contains the R2R3 repeat domains (note that the 295-311 segment that we model exists only if the isoform includes R2R3). We have now included a new paragraph into the relevant results section and incorporated the above discussion (please see the lines 251-263 in this new paragraph). We have also created a new figure, Supplementary Figure 9, which shows the alignment of these tauopathy-derived fibril structures onto the cross-beta turn of the xHAT monomer.

Our hairpin driven model presented in the paper is not restricted to the hairpin formation in the 295-311 tau segment. We focused on the 295-311 peptide fragments as a model system with mutant-sensitive hairpin propensity necessary for the xHAT mechanism. For different tauopathies, the xHAT mechanism can occur at different sequence locations depending on the isoform type, and possibly other factors such as post-translational modifications. This relates to the point raised by the referee about the absence of the heterosteric zipper in the tau residue segment of 295-311 in tau fibrils derived from AD patients. To our knowledge, this is because the characterized isoforms in all AD filament structures are missing the R2 domain that is part of the 295-311 sequence (see, for example 5o3l.pdb, 6hre.pdb, and 8bgs.pdb), which would make the xHAT mechanism unfavorable in this sequence region. However, the sequence composition of the turns in these structures are also consistent with the xHAT mechanism. For example, in all such AD fibril structures there is a central turn and heterosteric hydrophobic zipper around K343-K353. Our model suggests that these fibril structures may be seeded by a monomeric hairpin stabilized in this specific segment. In fact, close inspection of this region reveals an almost perfect alternating nature of polar/charged and hydrophobic residues. A hairpin with a turn having K347-D348-R349 charged residues can place all the remaining odd numbered residues (which are interestingly all polar/charged; K343, D345, Q351, and K353) on one face of the hairpin while leaving the even-numbered residues (all hydrophobic except S352; L344, F346, V350, and I354) on the other side. Such hairpins that are polar/charged on one side while hydrophobic on the other side are exactly what we observe in aggregation-prone mutants of 295-311 peptide segment in the current study, and biophysically very suitable for the pre-xHAT to xHAT transition. We have now added a discussion of this point on lines 263-271.

b. Nomenclature. 3D structures of filamentous tau aggregates from multiple diseases are now available, revealing architectures consisting of proto-filaments and filaments. Fibrils have been observed but only in vitro. In contrast, the manuscript uses the term “fibril” in a different sense and creates the term “protofibril” which has no established meaning. Authors should harmonize their nomenclature with the tau structure field so that readers can better understand their findings.

Agreed. We were using the term “protofibril” to mean a kind of linear form prior to a protofilament formation, but, as pointed out by the referee, we realized that this term doesn't have any established meaning. We have now replaced the term “protofibril” with “proto-filament” (line 13 and line 38), “linear dimer” (line 202-203), and “linear form” (line 215).

2. Experimental conditions need clarification

a. Authors work with 200 micromolar peptide, which is far above physiological tau concentrations. Authors should justify the conditions

We thank the reviewer for bringing up this point. In prior studies using similar peptides, we empirically established their aggregation capacity (Chen et al. 2019 Nat Comm, Chen et al. 2023 Nat Comm., Li et al. 2023 Structure). The overall rationale for using 200 uM concentrations of these fragments is that it corresponds roughly to 0.4mg/ml which is similar to the mg/ml of FL tau that is used in in-vitro aggregation studies. For the peptides, higher concentration is needed to form sufficient fibril surfaces that can bind to Thioflavin T above the fluorescence detection threshold. For this study, we also empirically compared aggregation capacity of WT and aggregation-prone P301L peptides at a range of concentrations (25, 50, 100, 200, 300 and 400uM) to determine sufficient amount of WT for robust fluorescence signal. Additionally, we tested seeded reactions using 100 uM monomer and the signal was variable, consistent with 200 uM monomer concentrations yielding more defined fluorescence amplitudes. WT and P301L peptide aggregation time series are now included in a new figure: Supplementary Figure S8.

b. Two constructs termed k18 and k19 are employed in the manuscript but not explicitly defined or referenced. Their nature is particularly confounding because deletion mutants named K18 and K19 have been used for many years in the tau field. Both K18 and K19 lack the C-terminal portion of the 4th microtubule binding repeat (as defined by sequence alignment), making them incomplete models of the microtubule binding repeat region. Authors should explicitly define k18 and k19 and explain why these specific constructs were chosen for experimentation.

Upon the suggestion of the referee, we have inserted the definition of k18 (one with R1R2R3R4 repeat domains), and k19 (having R1R2R3) into the supplementary Fig S1. We have also reformatted residue numberings of the 17 residue 3R fragment (in the bottom right of supplementary Fig S1) so that our mutant constructs are clearer. The main reason for our 17-residue mutants constructs, some from R2R3 and some from R1R3, is because we are comparing with kinetic aggregation data on these mutant peptides from our previous paper (Nat. Comm., 2019, Ref.27). For other simulations reporting on full repeat domains, we used the k18 construct because it has all four repeats, as indicated in the methods section. The reason for omitting C-terminal part of the microtubule binding domain is because the literature NMR data that we use to compare our predictions (which provides dominant beta-sheet regions) were performed on k18 and k19 constructs missing the C terminal part. As noted by the referee, k18 and k19 have been widely used as model systems for tau aggregation.

c. The depiction of parallel and anti-parallel beta sheets in Fig. 2a,b is unclear; an emphasis on them being intermolecular anti-parallel (or parallel) interactions would be appreciated by journal readers.

This point is well taken. We have made two changes in Fig2a and Fig2b to emphasize parallel and anti-parallel beta sheets. First, the top titles have been changed from “parallel” and “anti-parallel” to “intermolecular parallel” and “intermolecular anti-parallel”, respectively. Secondly, we have added orange -dashed lines indicating the

intermolecular hydrogen bonding to draw attention to intermolecular beta interactions, which are either parallel or anti-parallel.

3. New methodology has not been fully interrogated

a. Experimental verification of the simulation method (Fig. S2) leverages only two peptides of different lengths. The most accurate helix reproducing structure occurred only 33% of the time. It is not clear that novel methodology has been adequately vetted prior to its application to tau peptides.

The main methodology we used in folding simulations, Replica Exchange Molecular Dynamics (REMD) using Generalized Born Surface Area (GBSA) implicit solvent model with AMBER96 force-field, is not the novel part of our approach. Since its invention in the late 1990s, REMD has been a popular methodology applied to many different proteins/peptides folding simulations with many different force-fields and solvent models. Specifically, use of REMD with GBSA and Amber96 force-field (our methodology) were shown to be successful on a variety of proteins/peptides as we referenced in the manuscript (see lines 86-87 and 342-344). To go into some specifics, Ref. 36 (PNAS,2007) shows REMD/GBSA/Amber96 combination correctly folds 8 out of 9 selected small proteins, Ref. 37 (J. Chem. Phys., 2009) reports that combination of Amber96 force-field with implicit solvent models predicts stable structures that are in better agreement with experiments compared to other force-fields, Ref. 38 (Biophysical J., 2009) predicts correct folding of Fip35 Hpin1 WW domain using Amber96 in implicit solvent, and Ref. 39 (J. Phys. Chem. B, 2008) tests four different Amber force-fields in implicit solvent models using REMD, and reports that Amber96 is the best one having a well-balance between helix and sheet structures. In addition to these literature studies, we further tested the methodology on one alpha helical and one beta-sheet structures, and the methodology correctly predicts the NMR structure, as shown in Fig. S1. In this regard, the referee points out that our method only gave 33% of the NMR structure on alpha helix while hairpin one was predicted to be much better in 72%. First, as shown in Fig.S1, not only the first largest cluster (32.9%), but also the second (12.9%) and the third (7.2%) largest clusters are all partial helical folds of the complete fold. In fact, a hairpin fold of this helical peptide appears only in the 13th largest cluster, with only 1.4% probability. Overall, 82% of the conformations do have some sort of helical character while those with beta-sheets are less than 3%. Secondly, if we use a loose cut-off value of 5 Angstroms (instead of the conservative 3 Angstroms we used originally for both peptides), then 69.4% of the conformations turn out to be near-complete alpha helical folds, such that the median structure has an RMSD of 2.7 A with the NMR helix. We have **included these explanations into the related part in the supplementary (see lines 669-673).**

b. A justification for the xHAT score (lines 187-189) should be provided to ensure that it is not arbitrarily defined to improve fit.

The explanation for why we chose even-even-odd combinations for the xHAT mechanism appears at the end of the first paragraph in section III (please see lines 160-170).

Following the referee's comment, we realized that the reader may not find it straightforward to connect the first mention of the xHAT score to the justification of even-even-odd combination explained in the previous paragraph. Therefore, we have now **included a short sentence of "even-even-odd combination explained in the previous paragraph"** in parenthesis (lines 191-192), to refer the reader to the previous paragraph for a detailed explanation of the xHAT score.

Responses in **bold**, changes in the manuscript indicated with underlining.

Reviewer #2 (Remarks to the Author):

This manuscript presents important insights into the nucleation of fibril formation based on a combination of detailed computer simulations and cross-linking experiments. The cross-linking experiments are particularly strong evidence that the trimeric structure the authors have identified, referred to as the xHAT model, is the rate-limiting transition state for nucleation. The simulations were key for designing this experiment and help make sense of the data. The results are non-obvious, since a trimer that mimics the final structure of the fibril is insufficient to catalyze fibril formation.

We thank the Reviewer for appreciating the importance of the mechanistic insights provided by our interdisciplinary approach.

My major concern is with how the data are presented. The story, as told, is non-linear and makes jumps that are hard to follow. I almost gave into the temptation to write the results off and put the paper down at a couple of points. I don't think additional experiments are needed, but some significant restructuring of the manuscript/figures is warranted. Here are some of my concerns.

We thank the Reviewer for a careful reading despite the organization issues of the manuscript. We found the comments helpful and incisive, and have addressed them to the best of our ability. We believe the manuscript has improved significantly due to these changes. We are grateful for the reviewer's time and consideration.

- I wasn't clear why the authors started with simulations of the monomer, given that there is existing evidence an oligomer is the rate-limiting step and the authors seemed to agree with this conclusion. I think an appropriate hypothesis is that the oligomer is made up of energetically favorable monomer conformations, but neither this hypothesis or any competing hypothesis was ever given to motivate the initial work. Some hypothesis should be given for why the hairpin content of the monomer is predictive of the rate of fibril formation even though the rate-limiting step is a trimer.

We want to thank the Reviewer for pointing out such important points on the kinetics. Both to answer the referee's comments and to place our work into what is known on the amyloid kinetics, we now starting our discussion section with a newly added paragraph with new references (lines 293-302):

"In contrast to simple crystallization, previous studies suggest that amyloid fibril formation should be described by a multi-step nucleation mechanism in which oligomeric intermediates play crucial roles [51]. In particular, a widely accepted model called "Nucleated Conformational-Conversion (NCC)", formulated by two experimental [52, 53] and a later theoretical study [54], suggests that critical nuclei form by key structural conversions at some oligomeric state. More recent studies also support such an oligomer-driven fibril formation [55, 56, 57, 58]. In the NCC model, the nature of the rate-limiting step remains unresolved. Although it is known that an oligomer plays a central transition-state role the atomistic structure of such an oligomer remained elusive. Furthermore, it was unclear if the rate-determining step is from monomer to oligomer, or from oligomer to fibril. Here, our study proposes a well-defined trimeric hairpin complex

as the key oligomer structure, and predicts the rate-limiting step to be the formation of this oligomer from monomeric hairpin conformations.”

Regarding the Reviewer’s comment on the possible hypothesis that the oligomer is made up of energetically favorable monomer conformations, the xHAT model proposes important monomeric conformations that should have key structural features (beta sheet hairpins with shifted registry hydrogen bonding). However, these specific conformations may or may not be the energetically favorable ones. Although we do see that disease associated mutations significantly increase such hairpin populations in the monomer ensemble, they are still around 30% at most (see Table S1). Therefore, it might be the case that such conformations are in fact energetically unfavorable in native proteins, and external perturbations like mutations, PTMs, or other disease related biochemical environment elevate them to some significant levels which in turn promote the amyloid formation via the xHAT mechanism.

- Adding to the confusion, the authors group monomers into different registries without a clear rational/hypothesis for doing so.

As explained in lines 98 to 105, alternating polar and hydrophobic residues have already been known for amyloidogenic sequences. In our case, "even" hairpins are ones where even-numbered residues (which are dominantly hydrophobic; 306V, 308I, and 310Y) form hydrogen bonds to the N-terminal side, while polar/charged residues (305S, 307Q, and 309I) are internally hydrogen-bonded to N-terminal residues in "odd" hairpins (see Fig.1b). Therefore, in odd hairpins, hydrophobic residues expose their backbone to the solvent. Although, these explanations (with additional details) are already in lines 98 to 105, we realized that a few more sentences should be added to make it clearer. **Therefore, the following sentences have been appended to the explanation above: (lines 105 to 108)**

“Because amyloid fibrils have intermolecular backbone-to-backbone hydrogen bonding, solvent-exposed backbones of the hydrophobic residues can potentially promote amyloid formation in intermolecular interactions. To get more insight into the alternating sequence nature, and different possible roles of polar and hydrophobic residues, we quantified odd and even hairpin contents separately.”

Figure 1 also refers to the xHAT state, which is not even been defined in the first few sections of the manuscript (i.e. on the monomer and dimer). These data should either be motivated earlier on, or moved to later in the paper to help explain data on the trimer.

We agree with the referee that the plot for the correlation of xHAT score with the experiment (Fig1f in the original manuscript) should have been placed after the introduction of xHAT mechanism. The flow and the order of the discussion is correct but referencing Fig1f while discussing xHAT (Fig2) is not the best flow. Therefore, we have now moved Fig1f of the original manuscript to the new Fig2e.

- Similarly, there was not a clear rational/hypothesis motivating the work on the dimer. I think the authors meant to show that the dimer is not the rate-limiting step. But I didn’t feel like they started to address that until after talking about results on the dimer, leaving me confused. By the end of the discussion of the dimer, I was just confused what the authors were trying to show and

what I was supposed to take away from any of the data they presented.

The main reason for dimer simulations is that once we found monomeric hairpins correlate with the experimental rate, we first prepared dimers of such hairpins to search for any possible fibril-like cross-beta structural transition. Cross-beta structure cannot exist in monomer ensemble because both sides of the inner hydrophobic sidechains would have to face to polar solvent (and therefore no such monomeric cross-beta structure exists). Therefore, we must have fibril-like cross-beta structure at some oligomeric level. Although we did mention that we couldn't find any such transition at the end of the discussion on the dimer simulations, it is correct that we were missing the earlier motivation. Therefore, we have now rephrased the first two sentences of the related paragraph (lines 145-148) as:

“The correlation between hairpin content and fibril nucleation speed suggests that amyloid formation is somehow fed by monomeric hairpin motifs. To search for any fibril-like cross-beta structural transition, we prepared systems of aggregation-prone P301S mutant dimer and trimer complexes containing combinatorial arrangements of hairpins sampled from the monomer ensemble. First, starting from two hairpins,”

Reviewers' Comments:

Reviewer #1:

Remarks to the Author:

Major points - This reviewer's concerns have been addressed

Remaining minor concern:

1. The k18 construct is now defined as 243L - 375K on p. 7 line 220 but as 244-372 in Fig. S1. The discrepancy should be rectified. If the latter is correct, then the construct conforms to the original K18 produced by the Mandelkow lab. k19 is defined as 244-274/306-372 in Fig. S1, which does conform to the K19 construct from same lab. Once corrected, authors may consider citing PMID:16700555 for both k18 and k19.

Response to reviewer comments

Reviewer 1 only had one minor comment:

“The k18 construct is now defined as 243L - 375K on p. 7 line 220 but as 244-372 in Fig. S1. The discrepancy should be rectified. If the latter is correct, then the construct conforms to the original K18 produced by the Mandelkow lab. k19 is defined as 244-274/306-372 in Fig. S1, which does conform to the K19 construct from same lab. Once corrected, authors may consider citing PMID:16700555 for both k18 and k19.”

Response: We thank the reviewer for pointing this out. We have corrected the definition on line 216 (previously 220 before reduction of abstract length) to be 244-372. PMID:16700555 is now cited both in line 216 and in Fig.S1 caption.

Reviewer 2 had no comments